# Promoting Comfort: A Clinician Guide and Evidence-Based Skin Care Plan in the Prevention and Management of Radiation Dermatitis for Patients with Breast Cancer

**DOI:** 10.3390/healthcare10081496

**Published:** 2022-08-09

**Authors:** Deborah Witt Sherman, Sandra M. Walsh

**Affiliations:** Nicole Wertheim College of Nursing and Health Sciences, Florida International University, 11200 S. W. 8th St., Miami, FL 33199, USA

**Keywords:** breast cancer, skin care, radiation therapy, radiotherapy, radiation dermatitis, radiation skin reactions, prevention, management

## Abstract

Patients with breast cancer may be offered adjuvant radiation therapy (RT) after surgery. Up to 95% of these patients develop radiation dermatitis (RD) during or following RT. Randomized clinical trials and other literature provide evidence that RD can be prevented or reduced. The aim of this article is to propose a Clinician Guide and Evidence-based Skin Care Plan to prevent and/or reduce radiation dermatitis and promote the comfort of breast cancer patients receiving RT. As an integrative review, the databases searched were CINAHL and Medline, using the key terms: breast cancer, skin care, radiation, radiation therapy, radiotherapy, radiation dermatitis, and radiation skin reaction, prevention, and management. Search criteria included English language, full text, published between 2012 through 2020, and peer-reviewed. The search yielded 320 articles. Relevant articles were evaluated using the Quality Assessment Tool (QAT), and highly rated articles were selected to be included in the review of literature. The outcomes were the development of a Clinician Guide to offer holistic, patient-centered care and an Evidence-based Skin Care Plan. The research literature supports a standard skin care regimen, along with use of an emollient cream to the treatment area, use of deodorants depending on patient preferences, and application of a topical steroid cream daily throughout treatment and two weeks post RT. Clinician’s weekly assessments of patients offers therapeutic support and ensures optimal skin care during and post-RT. The comfort of breast cancer patients receiving RT requires the best level of evidence regarding the efficacy of interventions, coupled with clinician’s judgement, and patient’s preferences and wishes. The clinician-patient relationship is essential in addressing the physical, emotional, social, spiritual, and functional challenges associated with a cancer diagnosis and adjunctive radiation therapy to improve long-term survival.

## 1. Background

The American Cancer Society [1] estimated that in 2020, 276,480 women and 2620 men would be diagnosed with invasive breast cancer in the United States. Over a lifetime, one out of eight women in the United States will develop invasive breast cancer, which is the second most common type of cancer in women. Treatment options include mastectomy, lumpectomy, chemotherapy, hormonal therapy, and adjuvant radiation therapy (RT) [1,2].

According to Hogle [3], radiation treatments affect the skin’s anatomy and physiology depending on the dose, fraction size, volume of tissue, duration, energy and type of radiation, or bolus doses. The epidermis of the skin includes a cornified outer layer and a deeper basal layer. The epidermis is continuously renewed through a production of new skin cells from the basal layer in response to the shedding of skin cells at the outer layer. Radiation disrupts the balance between the production of new cells and the shedding of cells, resulting in mild to severe radiation dermatitis (RD). In addition, there are inflammatory responses with the release of histamine and serotonin, and vascular responses leading to capillary dilation in the dermis, which is the layer underneath the epidermis, in which there are nerve endings, blood vessels, and hair follicles. The skin responds to radiation with redness (erythema), changes in skin pigmentation, hair loss, and sweat and sebaceous gland destruction [4].

Varying degrees of RD occur within 1 to 4 weeks of radiation treatment and persist for 2 to 4 weeks following treatment. Transient erythema may occur within 24 h of treatment where the skin appears red, warm, and rash-like, and the patient experiences a sense of skin tightness and sensitivity. As the radiation doses increase to 20 Gy, the patient may experience dry desquamation, in which the skin becomes dry, itchy, or flaking. At doses of 30 to 40 Gy, extracapillary cell damage occurs with increasing edema. At doses 45 to 60 Gy, moist desquamation may occur in which the area may blister, bleed, slough, and ooze serous fluid with possible crusting [5].

For patients treated with RT, up to 95 percent are at risk for radiation dermatitis (RD) [6,7,8,9]. When RD progresses to acute radiation dermatitis (ARD), moist desquamation of the skin increases the possibility of infection [10,11,12].

Healthcare professionals often use a universal radiation therapy oncology group (RTOG) assessment tool to describe RD with a range of grades from 0–4. Rosenthal, Israilevich, and Moy [8] described the clinical presentation of RD based on the RTOG from grade 0 (normal skin) to grade 4 (ulceration and necrosis) which is the most acute dermatitis. However, the RTOG scale does not assess symptom severity such as pain. An additional assessment tool developed by the National Cancer Institute [13] is the Common Terminology Criteria for Adverse Events that rates the progressive severity of RD from grade 1 to grade 5, in which grade 1 is erythema or dry desquamation, while grades 2 and 3 represent moist desquamation with increasing discomfort and pain and grades 4 and 5 indicate skin ulceration and necrosis. Once RD begins, tissue damage builds with every subsequent radiation dose which further delays healing [14]. The consistent use of assessment tools is important to document the severity of RD and respond with appropriate therapeutic interventions. As it has been reported that RD grading by clinical assessments, such as the RTOG criteria, does not correlate well with patient-reported outcomes, there is a need for improved RD symptom assessment that includes both patient and clinician components [15].

Over the past 50 years, multiple risk factors have also been identified with RD, including individual/patient related factors and treatment related factors. While some of the risk factors are modifiable, others are not. Individual factors may include older age, smoking status, body mass index (BMI), type 2 diabetes mellitus, chronic immunosuppression, autoimmune disease, tumor histology and state, concurrent treatment with chemotherapy or hormonal therapy, compromised nutritional state, breast volume, initial darker skin, or chronic sun exposure [16,17,18]. Treatment factors may include whole breast fractionation schedule and dose, tumor bed boost dose, location of the tumor, duration of treatment, and type of energy used [9,16,17]. Newer radiation techniques may lower the incidence and severity of radiation skin reactions. For example, intensity-modulated radiation therapy (IMRT) results in only small volumes of normal tissue receiving the full treatment dose [5].

On a psychological level, patients receiving RT often express a loss of control, sleep disturbance, anxiety, depression, and issues with body image which are equally important to address as physical complications of treatment [19]. On a functional level, treatment related skin reactions may also lead to discomfort in wearing clothing and undergarments and in performing activities of daily living [20]. Prevention and reduction of RD are therefore extremely important. As RD progresses to severe ARD, not only is the condition very painful and debilitating, negatively impacting patient’s quality of life, but may result in patients’ or physicians’ decisions to terminate radiation treatment early or a patient’s decision not return for follow up appointments [10,11,12]. Beyond quality of life, a breast cancer patient’s survival from the disease is in significant jeopardy over time.

Clinicians’ knowledge regarding the assessment and management of skin reactions caused by radiation therapy is critical to promoting the comfort of breast cancer patients receiving RT. A holistic approach to care is important as radiation therapy impacts not only patients’ physical adjustment, but emotional and functional adjustment to cancer and its treatment. Throughout the course of RT, the goals of care include maintaining skin integrity, reduction of pain, protection from trauma, prevention and management of skin infections, and promoting a healing environment to the wound bed [5]. Beyond physical healing, it is important for clinicians to provide a healing environment in which the patient feels understood, listened to, and supported through a patient-centered approach throughout their treatment experience. This can be achieved by use of a stress-reduction approach during all interactions with patients, including guiding patients in self-care, reducing their stress, and promoting a healthy lifestyle during and following radiation treatment [21] while being mindful of the economic costs of treatment, access, and ease of following a skin care plan.

The purpose of this article is to propose a clinician guide and evidence-based skin care plan to prevent and/or reduce radiation dermatitis and promote the physical and emotional comfort of breast cancer patients during and following RT, based on the highest level of evidence regarding pharmacologic and non-pharmacologic treatments, as well as clinician judgment, and patient preferences and wishes.

## 2. Methods

An evidence-based approach to care is also important to identify the highest level of clinical evidence available to address a clinical problem such as RD. As an integrative review, the data bases searched were Medline and CINAHL. The key search terms entered were breast cancer, skin care, radiation, radiation therapy, radiotherapy, radiation dermatitis, radiation skin reactions, prevention, and management. Article selection criteria included: English language, full text, and published from 2012 to 2020 in peer reviewed journals. Systematic reviews, literature reviews, single randomized controlled trials, quality improvement projects, practice guidelines, and education-based articles were reviewed for relevance to the topic. The Medline search yielded 188 articles, and the CINHAL search yielded 132 articles. Duplicate articles across the two data bases were removed. Of the 320 articles identified, titles and abstracts were reviewed, and relevant articles were evaluated using the Quality Assessment Tool (QAT) developed by Sirriyeh, Lawton, Gardner, and Armitage [22], which has 14 evaluative criteria. Highly rated studies conducted both within the US and internationally are presented in the review of literature and were used as a basis for a clinician guide and evidence-based skin care plan to promote the comfort of breast cancer patients receiving RT.

## 3. Selected Research Findings

The development of evidence-based guidelines or protocols in the prevention and treatment of RD has been met with methodological challenges as studies vary in the treatments being compared, sample sizes, severity scoring, types of radiation therapy being offered, and the outcomes being measured, such as incidence and severity of RD, rather than delay to onset, progression, and duration. There is also a paucity of studies that focus on symptoms, interference with ADL, and quality of life. Yet, what has been learned through a review of the evidence has culminated in the best guidance and recommendations that can be made by clinicians at this point in time. The articles included in this literature review were evaluated as being of high quality to inform the development of a clinician guide and evidence-based skin care plan for breast cancer patients receiving RT including systematic reviews, clinical trials, quality improvement projects and clinically relevant papers presented in chronological order beginning with most recently published. While no gold standard for management of radiodermatitis exists, research is emerging and methodologies exist to synthesize evidence that inform decisions at the point of care [23].

Updating a previous systematic review to include articles published through September 2019, Ginex et al. [23] sought studies to inform the evidence-base for the Oncology Nursing Society Guidelines on the management of radiation dermatitis in patients with cancer. The authors presented the PRISMA flow diagram which initially identified 4376 articles. Based on the eligibility criteria, 22 articles additional articles were identified which addressed interventions to minimize or treat RD during cancer treatment. This review provides evidence of the efficacy of the following therapies in the prevention or reduction of RD: (1) use of aluminum or non-aluminum-containing deodorants causes no harm and may be used with a standing washing/skin care regimen in accordance with the values and preferences of the patient; (2) use of general emollient creams and lotions (i.e., Aquaphor, Sorbelene, Eucerin) to soften and moisturize the skin as a part of standing washing/skin care regimen; (3) use of semipermeable dressings (i.e., Mepitel film, StrataXRT, hydrofilm, and Mepilex), plus standard washing/skin care regimen; (4) use of a mid to high potency topical steroid (i.e., Hydrocortisone, Betamethasone, Fluticasone, Triamcinolone, Mometasone furoate, Clobetasol) which can be over-the-counter or prescribed, plus standard washing/skincare regimen; (5) use of a mid to high potency topical steroid for individuals with RD associated pain, burning, and itching; and (6) use of silver sulfadiazine (sulfa derivative topical antibacterial) for patients with moist desquamation. Aloe Vera and oral curcumin were recommended only in the context of a clinical trial. The results of the systematic review discouraged the use of the following treatments: (1) emu oil; (2) topical nonsteroidal interventions (i.e., Vitamin D ointment, Cavelon Barrier Cream, or oil- based emulsion containing allantoin; and (3) calendula (marigold plant).

In 2019, Karbasforoos et al. [24] conducted a clinical trial to study the use of Silymarin 1% gel (n = 20) vs. the control group (n = 20). The gel was applied once a day for the five weeks of RT. The RTOG severity scale was significantly lower in the Silymarin group as compared to the control group, with delayed development of RD after 5 weeks of application. This product is available on-line in the US as a cream called milk thistle, which is 2% Silymarin, but is stronger than the cream used in this study.

In 2018, Yee et al. [9] reviewed 96 randomized clinical trials (RCT’s) about radiation induced skin toxicity in breast cancer patients. Various factors were examined, such as radiation techniques, topical applications, oral supplements, and skin care protocols. Across trials, there was consistency in the use of the same assessment scales to determine skin toxicity established by the Radiation Therapy Oncology Group (RTOG) or the National Cancer Institute’s Common Terminology Criteria for Adverse Events (CTCAE) [9]. Yee et al. discussed the inconsistencies in the literature regarding factors that may predispose patients to RD with progression to ARD [16,17]. Conclusions indicated variation in results across studies, with no clear guidance about treatment. However, Yee et al. concluded that the use of different radiology techniques might decrease rates of RD. Additionally, Yee et al. noted there was limited investigation of patients’ satisfaction during RT and an absence of guidelines for RD management.

Furthermore in 2018, Lucas, Lacouture, Thompson, and Schneider [25] received approval from the ethics and medical review boards of Memorial Sloan Kettering Cancer Institute in New York City to conduct a quality improvement (QI) project with breast cancer patients who were receiving RT. The QI project was designed to improve clinicians’ knowledge about RD and increase the use of a steroid cream which was added to a skin care protocol. The skin care guidelines implemented during the QI project listed two major points about skin care: (1) washing the chest/breast with soap and water before each RT; and (2) the application of a mid to high potency steroid topical product to the radiated field following each RT and continuing for two weeks following the last RT. Physicians were encouraged to prescribe Mometasone, a topical steroid for prophylaxis. After implementation of the QI project that included the new skin care guidelines, patients’ charts were audited. Results indicated that the numbers of patients who were prescribed Mometasone increased from 13 to 77 patients, as prophylaxis for RD. While physicians did increase prescriptions for some type of topical application for prevention of RD, they prescribed 39 different products even though Mometasone was part of the new protocol. Those patients who were prescribed Mometasone (n = 50) had the fewest skin reactions during and following RT.

In 2017, Ulff et al. [26] examined the efficacy of Betamethasone, as a high potency steroid cream, in a group of 101 patients, as compared to a patient group of 101 who used Essex cream, which is a moisturizer. Patients applied the creams twice a day four hours before and directly after RT. The Betamethasone steroid cream significantly reduced RD as compared to Essex cream (moisturizer). The authors suggested that preventive application of a potent corticosteroid cream should be used routinely at the start of RT and continued for two weeks following RT.

In 2018, Haruna et al. [27] conducted a systematic review and meta-analysis regarding the topical management of acute radiation dermatitis in breast cancer patients. Ten randomized controlled trials involving 919 participants met the inclusion criteria. Meta-analysis, including 845 participants, demonstrated significant benefits of topical corticosteroids in reducing the mean RD severity score and preventing the incidence of RD progression to wet desquamation. Results further indicate that topical steroids decreased itching but not pain; however, steroids increased reported quality of life.

In 2016, Erridge et al. [28] conducted clinical trial with 219 patients regarding management of RD who either applied Betamethasone 0.1% or no cream once a day from the start of RT to two weeks post-RT. The results indicated that patients in the experimental group scored lower for itch, redness, and discomfort compared to those in the control group. Patients using Betamethasone also reported significantly lower sleep disturbance and use of analgesics.

In 2015, Kodiyan and Amber [11] reviewed 70 studies about skin care and various products tested for use with RD. They reported inconsistent findings regarding topical creams/ointments vs. placebos. They concluded that skin washing is recommended; anti-perspirants are allowed, and that topical corticosteroids may be useful.

Laffin et al. [29] also conducted a study in 2015 that examined the use of Cavilon barrier cream (n = 119) verses Sorbolene moisturizer (n = 126). Patients applied creams twice a day during treatment and for four weeks after treatment. The nurse made a follow-up call four weeks post RT. The results indicated that either product may be useful to prevent or lessen RD. Cavilon cream was more acceptable to patients, but Sorbolene was reported to better reduce symptoms such as itching. The authors mentioned that a steroid cream may be more beneficial than either product and discussed the importance of patient education to promote patient adherence to skin care guidelines.

As an additional study in 2015 regarding Cavilon barrier film compared to topical corticosteroid, Mometasone, Shaw et al. [30] studied 39 breast cancer patients who served as their own controls. Thirteen used Cavilon barrier film verses no treatment, 17 used topical Mometasone verses Cavilon film, while 9 used Mometasone vs. no treatment. The barrier film or topical steroid was applied every other day excluding weekends. The results indicate that patients using the Cavilon barrier film experienced a later occurrence of pruritus compared to the topical corticosteroid. However, the authors concluded that although not statistically significant both delayed occurrence of RD. If RD appears, it was suggested that the application of a corticosteroid be administered.

As early as 2013, the Multinational Association for Supportive Care in Cancer (MASCC) published practice guidelines to prevent and/or treat RD [31]. It was recommended that patients care for their skin from the beginning of RT until several weeks after RT by: (1) gentle washing with or without a mild soap; (2) allowing deodorant use; and (3) application of a topical steroid as prophylaxis for RD throughout and for several weeks post RT. These finding and recommendations are consistent with Lucas et al. [25].

In a study conducted by Hemati and colleagues [32] in 2012, 51 patients received topical silver sulfadiazine 1% as compared to the usual care/control group. During five weeks of RT for five days a week, patients applied topical silver sulfadiazine three times a day when receiving treatment and after RT applied the cream daily for one week. The results indicated that topical silver sulfadiazine significantly decreased skin injury as compared to usual skin care alone which was part of the treatment plan for both the intervention and control group. However, the authors suggested that patients be studied longitudinally to determine differences after the last week of medication administration.

In 2011, McQuestion [5] presented a clinical update of evidence-based skin care management in radiation therapy, noting that over the previous four years, there was minimal change in the evidence available to guide treatment decisions in the management of radiation skin reactions. Based on a review of the literature, the following recommendations were made: washing the site with lukewarm water and mild soap as a routine part of care; no harm caused by the use of metallic or non-metallic deodorants, given that magnesium, aluminum, or zinc did not cause an increase in skin reactions. There were mixed results of studies regarding the use of aloe vera, calendula, sucralfate, hyaluronic acid, topical steroids, or use of barrier films; while no benefit was seen with the use of biaffine (trolamine), which is an oil in water emulsion.

Additional important points made by Fenton-Kerimian et al. [33] in the care of patients receiving RT are the importance of weekly assessments during the course of treatment and follow-up post-treatment at one week, one month, and three months, as well as the importance of longitudinal studies to determine long term outcomes of patients with breast cancer who received RT.

Bauer, Laszewski, and Magnan [6] emphasized not only the need for patient education, but discussed the importance of patient’s active involvement in skin care during RT. They emphasized the role of the clinician in encouraging patients to adhere to a skin care protocol during RT and reporting any changes in skin integrity for the purpose of prevention and early treatment. Using the McCarthy’s 4MAT Model, Bauer et al. [6] recommended multiple approaches to engage patients in self-care, as well as assessment of patients’ different learning styles, each of which increase adherence to treatment plans and patient satisfaction with care.

Overall, the review of the literature supports a standard skin care regimen, along with use of an emollient cream to the treatment area, use of deodorants depending on patient preferences, and application of a topical steroid cream daily throughout treatment and two weeks post RT. Clinician’s weekly assessments of patients offers therapeutic support and ensures optimal skin care during and post-RT.

## 4. Translation of Evidence into Practice: The Development of a Clinician Guide and Evidence-Based Skin Care Plan

The literature review indicates that clinicians caring for breast cancer patients receiving RT are positioned to promote the translation of research results into practice. Based on the review of the literature and coupled with clinical judgment, the following Clinician Guide (Box 1), and Evidence-based Skin Care Plan (Box 2) are proposed in the care of breast cancer patients receiving RT.

Box 1A Clinician Guide to Promoting Comfort of Breast Cancer Patients Receiving RT.
**Week 1 (First Visit)**


**Build a Trusting Patient-Clinician Relationship:**
○Introduce the patient to members of the radiation therapy team and provide contact information to respond to questions.○Express the value of a holistic, patient-centered care and the importance of individualizing the approach to care dependent on their values and preferences.○Determine patients’ individual learning needs and styles and provide additional information in the format preferred, such as written materials, videos, possible Apps, and psycho-educational information through consultation with staff during each visit.○Encourage open communication with the clinician and health team members.○Discuss the patient’s physical, emotional, social, spiritual, and functional adjustment to the illness and the treatment.

**Provide Emotional Care and Support:**
○Assess and address patient’s fears and concerns regarding cancer and RT, with a reassuring, supportive approach.○Check-in with the patient following the first treatment to offer reassurance and engage them in their care. Development of a trusting provider-patient relationship fosters open communication and a positive clinical experience. 

**Promote Social Support:**
○Determine the need for additional support of family or friends in coming and returning home from treatments.○Discuss how treatments will occur within the context of lifestyle and other possible roles and responsibilities.

**Educate the Patient:**
○Discuss the value of radiation therapy (RT), how RT works in destroying cancer cells, skin changes associated with RT, when skin reactions may start, and the importance of protecting healthy cells and tissues in the affected area.○Review the expected procedures that will occur during radiation therapy to reduce uncertainty and promote emotional comfort.○Inform the patient that the radiation therapy team will assess any effects of radiation therapy through use of standardized assessment tools, which will be compared with patient-reported symptoms, with the goal of prevention and early management of radiation skin reactions.○Provide a step-by-step review of the Skin Care Plan (refer to Box 2) to prevent and manage RD and promote comfort.


**Week 2 to Week 5 (Second and Subsequent Visits)**


**Conduct a Physical Assessment:**
○At each visit by consistent staff, if feasible, to determine changes in the skin color, texture, moisture, and breaches in skin integrity, as well as assessment of signs of infection, and evaluation of symptoms such as pain, itching, sleep disturbance, anxiety, depression, and concerns about body image. 

**Provide holistic care:**
○Evaluate the patients’ overall adjustment to the illness and treatment.○Discuss positive coping strategies within the context of the illness and treatment experience.

**Encourage patient engagement:**
○Remind the patient to perform daily skin checks and record daily skin care changes and practices. A diary may be suggested. ○Review the diary at each visit to address patients’ individual questions and concerns, while attempting to normalize the experience, yet providing support and reassurance.○Continue to review the Skin Care Plan, as described in Box 2.○Encourage self-care, which may include the use of a guided imagery, relaxation techniques, hobbies of interest, use of distraction such as music, positive self-affirmations, prayer, need for additional professional support, support from family and friends, or in support groups with others who have successfully completed RT. Each strategy may increase a sense of emotional and spiritual well-being and promote comfort.○Reinforce healthy lifestyles, including adequate nutrition and hydration, sleep, and avoidance of tobacco, alcohol or other substances.



Box 2Comfort Guidelines: Evidence-based Skin Care Plan to Follow During and After Radiation.During radiation therapy, many people experience a skin reaction called radiation dermatitis (RD) ranging from slight to severe. Our goal is to work with you to protect your skin during RT and improve your comfort.
**FOLLOW THIS SKIN CARE PLAN THROUGHOUT TREATMENT AND FOR TWO WEEKS FOLLOWING TREATMENT**

A clinician will see you weekly or more often if needed. For immediate assistance between visits, you can receive help 24/7 by calling (name and/or phone number). _____________________________Keep a daily diary of skin changes/reactions to be shared with your radiation team members at each visit.
Skin red or pink color    ___     Areas that blister, weep, or peel ___Tanned color of skin   ___     Signs of crusting        ___Dry, itching, or flaking    ___      Signs of ulceration       ___Tender to touch       ___      Exudate/Discharge       ___Decrease in sweat      ___      Blackening of the skin      ___
Report symptoms of pain, burning, or itching so that your clinician can prescribe oral medications to alleviate symptoms and promote your comfort.
**Please follow the directions below to prevent or lessen radiation dermatitis.**


Protect the skin in the treatment area from sun and cold.Do NOT use hot packs, cold packs, or heating pads on the treatment area.DO NOT take baths, use hot tubs, or swim in lakes or pools if your skin is not intact.Wear soft, loose comfortable cotton clothing. Avoid underwire bras during the remainder of treatment.Do not rub or scratch the skin in the treatment area. Avoid shaving the armpit with a straight razor. May use an electric razor or do not shave if preferred.Perform Standard Washing and Skin Care: Shower before each treatment with a mild unscented soap (i.e., Dove, Neutrogena, or baby soap) and warm water.
▪Wash affected area and gently remove the skin product and deodorant during the shower. Do NOT scrub. ▪Dry treatment area with a clean, soft towel. Gently pat dry.▪Apply an emollient cream, such as Aquaphor or Eucerin, to moisturize the skin in the treated area following a shower.
You may use a non-metallic or metallic deodorants/antiperspirants as they promote comfort and do not cause harm. Use of deodorants is based on your preference.From the day of your first treatment until two weeks after treatment, apply a thin layer of mid to high potency topical steroid cream (e.g. Over the Counter: Hydrocortisone 1% (twice a day); Prescription: Betamethasone 0.1% (once or twice a day); Fluticasone 0.05% (twice a day), Triamcinolone 0.1% (twice a day), Mometasone furoate 0.1% (once a day), Clobetasol 0.05% (twice a day) to the radiation area after treatment. (Over the counter or prescription steroid creams may be used). NOTE: When using a topical steroid, apply moisturizer after the topical steroid. Use topical steroids only on intact skin.Speak with your clinician if your skin is NOT intact for additional skin treatments.Use no other skin care product on the irradiated area throughout treatment, including perfume or make-up.Avoid the use of tape and adhesives in the treatment area.Realize that fatigue may occur during radiation treatment; however, report to your clinician signs of systemic illness, such as fever, chills, or generalized weakness.Eat a healthy well-balanced diet to promote skin healing and increase your energy.Discuss with your clinician any physical, emotional, social, spiritual or functional issues you are experiencing.Make notes below as a reminder of issues to discuss with your clinician.


## 5. Implications Regarding the Use of the Clinician Guide and Evidence-Based Skin Care Plan in Improving Patient Well-Being and Outcomes during and following RT

A holistic, patient-centered approach by an interprofessional team of clinicians, as suggested in the Clinician Guide, provides patients with reassurance that they are receiving personalized, quality care based on expert knowledge [19]. The Clinician Guide educates practitioners on the cancer experience not only from a treatment perspective, but the emotional, social, spiritual, and functional implications related to RT. With education about RT, how it works, anticipated side effects, protective skin care practices, and ways of preventing and managing mild to severe skin reactions, patient’s fear and uncertainty would be reduced. Developing a trusting relationship with clinicians, enables patients to more fully process the experience, confront the challenges, and draw upon healthy coping strategies. Understanding the time-frame of treatment and the healing process post RT assists patients in setting realistic expectations regarding moving forward and resuming life beyond cancer and its treatments. Although members of the radiation team are not psychiatric counselors, getting to know the patient as a person, asking about their life, problems and challenges offers patients a sense of care and support beyond the technical aspects of cancer treatment. Education, support, reassurance, and kindness are therapeutic modalities that promote comfort and healing, and can be extremely impactful in successful coping during cancer treatment.

The weekly assessment of the treatment area by clinicians, based on objective RD assessment methods, along with a discussion of patient reported symptoms and a review of diary inputs by patients is valuable in determining a delay or progression of RD. Weekly review of the evidence-based skin care plan reinforces education regarding strategies to prevent or manage RD. Understanding how to care for the skin, prevent further skin damage, address skin reactions, and decrease symptoms empowers patients in terms of engagement in self-care and fosters a sense of control throughout the treatment experience. The stress of a cancer diagnosis and treatment can be overwhelming, along with day-to-day assaults when usual routines are disrupted. Yet, clinicians, as practitioners, educators, counselors, and advocates, can help make the experience bearable, avoid treatment delays or discontinuation, with the potential to enhance patients’ quality of life.

## 6. Future Research and Evaluation of Patient Outcomes

Randomized controlled trials are the gold standard for identifying the effectiveness of interventions. Systematic reviews synthesize the existing findings relevant to the prevention and treatment of RD, providing the best level of evidence available upon which to make clinical decisions regarding treatments. As more is learned about RD, the underlying pathophysiology and severity in relation to treatment dosing, associated risk factors, and potential mediating or moderating factors, there is an opportunity to test current and new pharmacologic, non-pharmacologic, and even complementary therapies (i.e., music, relaxation) to prevent, minimize, and treat RD. Clinicians need to be able to critique the research literature and evaluate studies to inform their clinical decision making and the development of evidence-based RD Clinical Guidelines or Protocols.

In addition to the physiological aspects of RT, additional research needs to be given to the emotional, social, spiritual, and functional challenges associated with RT. The review of the literature identified Kolcaba’s [34] Radiation Therapy Comfort Questionnaire designed for patients with breast cancer receiving RT. The use of the comfort questionnaire provides a holistic assessment of patients and their experiences during RT, assisting clinicians to not only to evaluate patients’ response to RT, but also to identify patients in need of further physical, emotional or spiritual interventions. Research studies, using a pre and post-test design, may evaluate patient’s knowledge regarding RD in relation to an educational intervention. Surveys may also be conducted to determine patient experiences to add to the evidence-base regarding skin care of breast cancer patients receiving RT. Qualitative studies, using a phenomenological approach, would be of value in examining the lived experience of breast cancer patients receiving RT and the meaning associated with RD and its treatment. Grounded theory studies may also reveal key concepts that may increase knowledge of risk factors, coping strategies, and overall impact of RT in terms of the patient’s quality of life. Equally important to evaluating short-term outcomes is the importance of conducting longitudinal studies that examine the long-term sequela associated with RT. Short-term and long-term interventions may be needed to promote a patient’s quality of life and protect their survivorship status.

## 7. Conclusions

The importance of holistic, patient-centered assessments, interventions, and education during RT should not be understated in its value of promoting patients’ physical and emotional comfort. The review of literature, summarized in this article, provides evidence that guides clinical practice, informing both the Clinician Guide to care, and the Evidence-Based Skin Care Plan which can be shared with patients at the onset of treatment and accessible to the community through institutional websites and patient education materials. The implementation of a skin care plan is an opportunity for patients to fully engage in self-care, not only promoting their skin health, reducing RT-associated side effects, and promoting the restoration of skin integrity, but can enhance their sense of control with the stressful context of cancer treatments. Ultimately, a patient-centered approach with implementation of a skin care plan may avert a delay in treatment or discontinuation of RT due to RD and afford breast cancer patients the greatest chance for long-term survival.

## Data Availability

Not applicable.

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
