# Peer review of "Promoting Comfort: A Clinician Guide and Evidence-Based Skin Care Plan in the Prevention and Management of Radiation Dermatitis for Patients with Breast Cancer"

_healthcare, 2022, doi:10.3390/healthcare10081496_

Round 1

Reviewer 1 Report

The manuscript has been improved for readability.

Reviewer 2 Report

The reviewer thanks the author for revising the manuscript. It became much more scientific, therefore, the reviewer deems it suitable for publication in Healthcare.

This manuscript is a resubmission of an earlier submission. The following is a list of the peer review reports and author responses from that submission.

Round 1

Reviewer 1 Report

The manuscript is generally well written.

However, there are some concerns.

title; The manuscript is a case report and a review of past studies. The title must be accurate.

I don't understand why the review manuscript requires a case report to be presented. The readers will be confused.

Case study:  It is not a Case study. This is a case report. The authors should clarify the need for case reports. Also, I suggest shortening case study including paragraph6.

The authors have fully evaluated past papers, but the inclusion of case reports masks important points in the manuscript. The authors need to understand that the manuscript is difficult to read due to with case report.

Reviewer 2 Report

The authors provide a narrative overview of different prevention and treatment options for radiation dermatitis in breast cancer.   I have several major remarks: - The way the manuscript is written does not really make it suitable for publication in a medical journal, even if the scope of the journal is healthcare. The important "Clinician Guide" is somewhat lacking and disorganised, not readily translating into everyday clinical practice and thus adding only little value to the scientific community. - Section 4 is disorganised, hard to read through and unstructured. A more fitting structure would be to describe general findings separately (e.g. skin washing, use of corticosteroids, use of deodorant) and then mention what author X, Y, and Z have written about these points. - Although sometimes useful in clinical publications, I do not see any added value of the case study in Section 2. Furthermore, the views stated in Section 6 are a somewhat idealistic projection ("I am a survivor") and lack foundation. If the case study cannot be omitted, consider moving it to the Supplements (and then also discuss the potential role of lymphoedema on the reduced arm mobility described by the patient). - Topical corticosteroids are useful for preventing and treating RD, yet widespread use remains limited due to the associated side effect profile (e.g. https://doi.org/10.21873/anticanres.11960). Please discuss. Regarding this, the respective active ingredients and their potency, dose and frequency, and mode of application should be better specified (especially in a clinical guideline). - Subjective RD grading by clinical assessments such as RTOG or CTCAE criteria do not correlate well with patient-reported outcome (e.g. https://doi.org/10.1016/j.radonc.2021.03.020). Furthermore, objective RD assessment methods, which are often not used, are available (e.g. https://doi.org/10.1016/j.radonc.2020.02.018, https://doi.org/10.1007/s00520-020-05889-w, https://doi.org/10.3390/polym11122112, https://doi.org/10.1007/s00520-018-4487-4). An RD toxicity assessment model including objective assessment methods as well as patient-reported outcome is needed. Please discuss (also under limitations). - It is unclear why US-based and international publications are discussed separately. The 7 publications in Table 1 are chosen somewhat arbitrarily (especially as the basis for an evidence-based clinical guideline). Please provide the exact methodology for selecting these papers under Section 3. - What are the limitations of the study? Which questions remain unanswered? Please discuss thoroughly. - The conclusion in Section 8 does not fit in with the rest of the manuscript.    Minor remarks: - Add breast volume and initial darker skin as risk factors for RD development (https://doi.org/10.3390/cancers12092444). - Check grammar and spelling throughout the entire manuscript (e.g. lines 57, 73, 136, 275, 306, 323, 329). - The terms "radiologist" and "radiation oncologist" are used somewhat arbitrarily. Please be consistent and correct. - Line 96: ... result in patients' or physicians' decision to pause or terminate radiation treatment ...   Overall, this manuscript cannot be considered for publication in its current form.